**⬤eLife**

# Automated cell annotation in multi-cell images using an improved CRF_ ID algorithm

Hyun Jee Lee[1†], Jingting Liang[2†], Shivesh Chaudhary[1], Sihoon Moon[1], Zikai Yu[3], Taihong Wu[2], He Liu[2‡], Myung-Kyu Choi[2], Yun Zhang[2,4*], Hang Lu[1,3*]

[1]School of Chemical & Biomolecular Engineering, Georgia Institute of Technology, Atlanta, United States; [2]Department of Organismic and Evolutionary Biology, Harvard University, Cambridge, United States; [3]Interdisciplinary BioEngineering Program, Georgia Institute of Technology, Atlanta, United States; [4]Center for Brain Science, Harvard University, Cambridge, United States

**\*For correspondence:**
yzhang@oeb.harvard.edu (YZ);
hang.lu@gatech.edu (HL)

[†]These authors contributed equally to this work

**Present address:** [‡]Advanced Institute of Natural Sciences, Beijing Normal University, Zhuhai, China

**Competing interest:** The authors declare that no competing interests exist.

## eLife Assessment

This Research Advance describes a **valuable** image analysis method to identify individual neurons within a population of fluorescently labeled cells in the nematode *C. elegans.* The findings are **solid** and the method succeeds to identify cells with high precision. The method will be be of interest to the *C. elegans* research community.

## Abstract

Cell identification is an important yet difficult process in data analysis of biological images. Previously, we developed an automated cell identification method called CRF_ID and demonstrated its high performance in *Caenorhabditis elegans* whole-brain images (Chaudhary et al., 2021). However, because the method was optimized for whole-brain imaging, comparable performance could not be guaranteed for application in commonly used *C. elegans* multi-cell images that display a subpopulation of cells. Here, we present an advancement, CRF_ID 2.0, that expands the generalizability of the method to multi-cell imaging beyond whole-brain imaging. To illustrate the application of the advance, we show the characterization of CRF_ID 2.0 in multi-cell imaging and cell-specific gene expression analysis in *C. elegans*. This work demonstrates that high-accuracy automated cell annotation in multi-cell imaging can expedite cell identification and reduce its subjectivity in *C. elegans* and potentially other biological images of various origins.

## Introduction

One of the bottlenecks in biological research is the inefficiency and inaccuracy of analyzing bioimages, which have become essential research materials owing to the emergence of advanced microscopy and imaging modalities (*Xu and Jackson, 2019*). For studies with the popular model organism *Caenorhabditis elegans*, one of the most challenging image analysis processes is cell identification: annotating the cell types in the image based on their anatomical or other biological features. Accurate cell identification is important in applications, such as gene expression analysis and calcium imaging, in order to obtain cell-specific information from multiple animals sampled in the experiment and associate this information with existing knowledge about the cell. Previously, we developed an automated cell identification method called CRF_ID based on graphical optimization using the Conditional Random Fields (CRF) model (*Chaudhary et al., 2021*). We demonstrated that for whole-brain images CRF_ID

shows higher annotation accuracy and more robustness against various sources of noise compared to conventional registration-based methods.

However, because CRF_ID was optimized for whole-brain images, it is not ideal for multi-cell imaging, which focuses on a subpopulation of cells. In fact, there are no automated methods currently designed for processing multi-cell images; only ad hoc and heuristic tracking and annotation methods are available. This represents an unmet demand because multi-cell imaging is still much more frequently used than whole-brain imaging despite the recent popularization of brain-wide imaging. For instance, multi-cell imaging is necessary for transcriptional or translational reporter-based gene expression analysis because the number of cells imaged is governed by the gene expression itself, with the average being around 40 neurons (*Taylor et al., 2021*). Another reason for the continued prominence of multi-cell imaging is that many biological questions can be effectively answered with a subset of the nervous system with specific structures and functions. For example, studies focusing on chemosensory neurons of the olfactory circuit in *C. elegans* characterized how the identity and intensity of olfactory stimuli are represented by the activity of the sensory neurons, which provided input signals for computation and integration of the downstream circuits (*Lin et al., 2023*). Furthermore, multi-cell imaging is more accessible than whole-brain imaging because it does not require fast-speed and multi-color volumetric microscopy techniques.

In this work, we present CRF_ID 2.0, an update of our original CRF_ID algorithm for multi-cell images. Compared with other automated whole-brain annotation methods (*Toyoshima et al., 2020*; *Yu et al., 2021*), CRF_ID is an ideal method to adapt for multi-cell images because of its demonstrated high accuracy, modularity, and efficiency for atlas-building; CRF_ID builds structured models rather than deep-learning models, making it readily interpretable. In principle, the original CRF_ID algorithm should be applicable to multi-cell images; however, in practice, multi-cell-specific modifications were necessary to achieve the highest accuracy. In addition to optimizing the method, we characterized its performance by comparing the accuracy of different types of atlases against each other and against manual annotations. These characterizations, which were not addressed in our previous work, provide a necessary reference for future users. Furthermore, we demonstrate the application of multi-cell neuron annotation in a cell-specific gene expression analysis. Thus, this follow-up work enhances the generalizability and usefulness of the CRF_ID method by enabling high-performance operation regardless of the number of cells in the images.

## Results

### CRF_ID 2.0 for automatic cell annotation in multi-cell images

The multi-cell identification pipeline is a multi-step optimization algorithm (*Figure 1a*), built upon CRF_ID (*Chaudhary et al., 2021*). First, the user acquires a volumetric image set of a sample containing multiple cells labeled with markers, such as fluorescent proteins or dyes (*Figure 1a1*). Image processing begins with cell segmentation. Here, we use a simple automatic method that identifies the local maxima of fluorescence intensity and fits the 3D Gaussian mixture model on them (*Figure 1a2*). Then, the coordinate axes of *C. elegans* body orientation are either assigned by the user or automatically predicted on the point cloud using an improved algorithm included in CRF_ID 2.0 (*Figure 1a3*). Next, based on the cell coordinates relative to the axes, the algorithm extracts the positional features of the cells, such as the pair-wise 3D positional relationships and angular relationships with other cells (*Figure 1a4*). Lastly, the CRF model (*Lafferty et al., 2001*) compares the extracted features from the dataset against a reference atlas, which could be derived from the literature or new data (*Figure 1a5*). The model computes a conditional joint probability distribution over all feasible cell identification (cell ID) assignments, and the neuronal cell ID assignments are ranked for each cell based on the computed probabilities (*Figure 1a6*). Note that a truncated candidate list can be used for subset-specific cell ID if the neuronal expression is known. Additionally, to maximize the accuracy of neuron identification, the reference atlas may be constructed by the user with use-specific datasets (*Figure 1b*). The atlas-building process is computationally simple and fast upon the availability of ground-truth datasets, which can be manually annotated by the user.

To demonstrate the utility of the automatic cell annotation algorithm for multi-cell images, we chose as an example a *C. elegans* strain carrying a red fluorescent protein, mCherry, expressed by a *glr-1* promoter (see the schematic and fluorescence images in *Figure 1*). The fluorescent protein

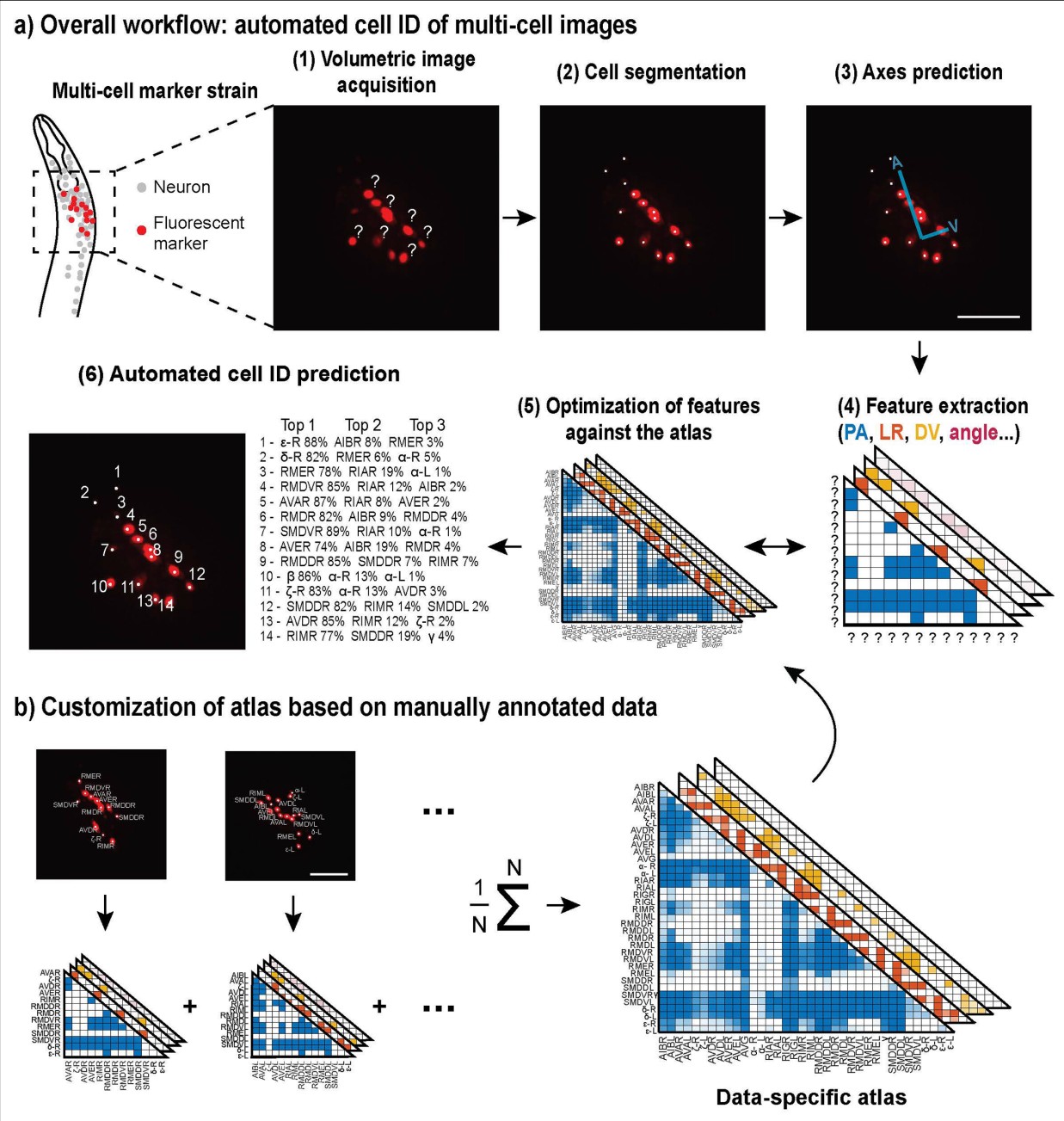

**Figure 1.** CRF_ID for multi-cell images. (**a**) Computational workflow starting from image acquisition to final cell identity predictions. (**1–3**) Image preprocessing steps include automatic cell segmentation and coordinate axes prediction. (**4**) Feature variables that represent positional relationships of the cells are extracted (PA, posterior and anterior; LR, left and right; DV, dorsal and ventral). (**5**) The Conditional Random Fields (CRF) algorithm maximizes the similarities between the extracted features from the images and those from an atlas. (**6**) The final results are represented as a list of most likely neuron candidates for each cell with predicted probabilities. (**b**) The atlas can be customized to meet the specifications of the images, and this is easily done by compiling and averaging annotated data. The images are showing half volume (left or right side) of the specimen for illustration. Scale bar: 25 μm.

was tagged with nuclear localization sequences (NLS) to confine the signal to the cell nucleus and aid in the separation of labeled neurons. *glr-1* is expressed in approximately 28 neuronal cell types with the majority localized in the head (*Brockie et al., 2001*; *Hart et al., 1995*; *Maricq et al., 1995*; *Taylor et al., 2021*). The set of neurons includes interneurons and motor neurons implicated in various behaviors and neuronal functions, including locomotion, chemotaxis, learning, and memory (*Gray et al., 2005*; *Hart et al., 1995*; *Liu and Zhang, 2020* and the references therein; *Maricq et al., 1995*).

This scale of expression is typical of many neuronal genes for most of which robust and easy cell identification methods do not currently exist. For this work, we collected images from transgenic worms expressing *glr-1p::NLS-mCherry-NLS* transgene and manually annotated 26 volumes to assess the performance of our method.

## New features improve prediction accuracy of body axes

While CRF_ID performs very well with whole-brain image datasets and has a great potential for generalization, multi-cell imaging poses challenges that require the algorithm to consider additional features in the data to accurately predict neuron identities. One such challenge is in the prediction of body axes from the volumetric images. It is important to correctly assign the three-dimensional coordinate axes anterior-posterior (AP), left-right (LR), and dorsal-ventral (DV) for each worm as this standardizes neuronal positions of worms imaged in various orientations. However, the axes assignment is not a trivial task because the worms, especially those in microfluidic devices, can deviate from its natural orientation and be at an angle as large as 20° from the plane (*Figure 2—figure supplement 1*).

In our previous work, principal components analysis (PCA) was employed on point clouds of head neurons segmented from fluorescence volumes (*Chaudhary et al., 2021*). Since PCA finds orthogonal dimensions that explain the most variance in the point cloud, the first three principal components would correspond to the coordinate axes of the worm, assuming the point cloud adequately represents the worm head shape and radial asymmetry of the nervous system. However, using PCA on cell point clouds to predict the coordinate axes is not suitable for multi-cell images. In whole-brain images, nearly all head neurons are fluorescent, which means the point cloud of the neurons is a fair representation of the worm's overall head shape. In contrast, multi-cell images have a smaller number of fluorescent cells, whose locations may not properly sample the space of the whole brain. For instance, if the fluorescent cells are concentrated near the ventral side of the head, the resulting AP axis would gravitate toward the ventral side, deviating from the ground truth.

To address the axes prediction challenge in multi-cell images, we have amended the method to be less dependent on the point cloud of cell centroids, which varies depending on which neurons are expressing the fluorophore. The new coordinate assignment method takes advantage of two common features in almost all samples—that the worm is autofluorescent, and that many neuronal pairs are bilaterally symmetric in their anatomical positions. It involves two correction steps (*Figure 2*). The first step corrects the AP axis by incorporating auto-fluorescence signals as natural landmarks to enlarge the point cloud (*Figure 2a*). This is easily implemented by imaging a volume in the green channel, where the autofluorescence is discernible and segmenting the fluorescent signals as points using the same cell segmentation method. The new point cloud then reflects the overall shape of the head, and the resulting AP axis from PCA aligns correctly along the head of the animal. The second step corrects the LR axis, for which we have implemented an algorithm that searches for the best plane of bilateral symmetry (*Figure 2a*). Using the initial LR axis as the starting point and the orthogonality to the AP axis as a constraint, the algorithm iteratively finds planes within a range and computes a symmetry score for the point cloud with respect to each plane. The plane that results in the highest symmetry score is assigned as the final LR axis. The DV axis is automatically determined by orthogonality to the first two axes.

In order to quantitatively evaluate the axes prediction performance, we manually defined 'ground truth' axes for each volume and calculated the angle deviations of the predicted axes (*Figure 2a*). Compared to the axes predicted by PCA on cell point clouds, the new axes that have been corrected by the two-step method all showed decreased deviations from the manually defined axes. More than 90% of the corrected axes were within 10° from the ground truth, which was comparable to the standard deviation of manual annotations in defining the ground truth. More importantly, the axes correction led to a significant improvement in the accuracy of the neuron ID prediction, measured using correspondence to human annotations (*Figure 2c*). Also, there was no significant difference between the corrected axes and the manually defined axes in terms of the resulting neuron ID correspondence, indicating that the automatically predicted axes are comparable to those defined by human. The details of the quantification of neuron ID accuracy are discussed in 'Materials and methods' under the section 'CRF_ID 2.0: evaluation of accuracy'.

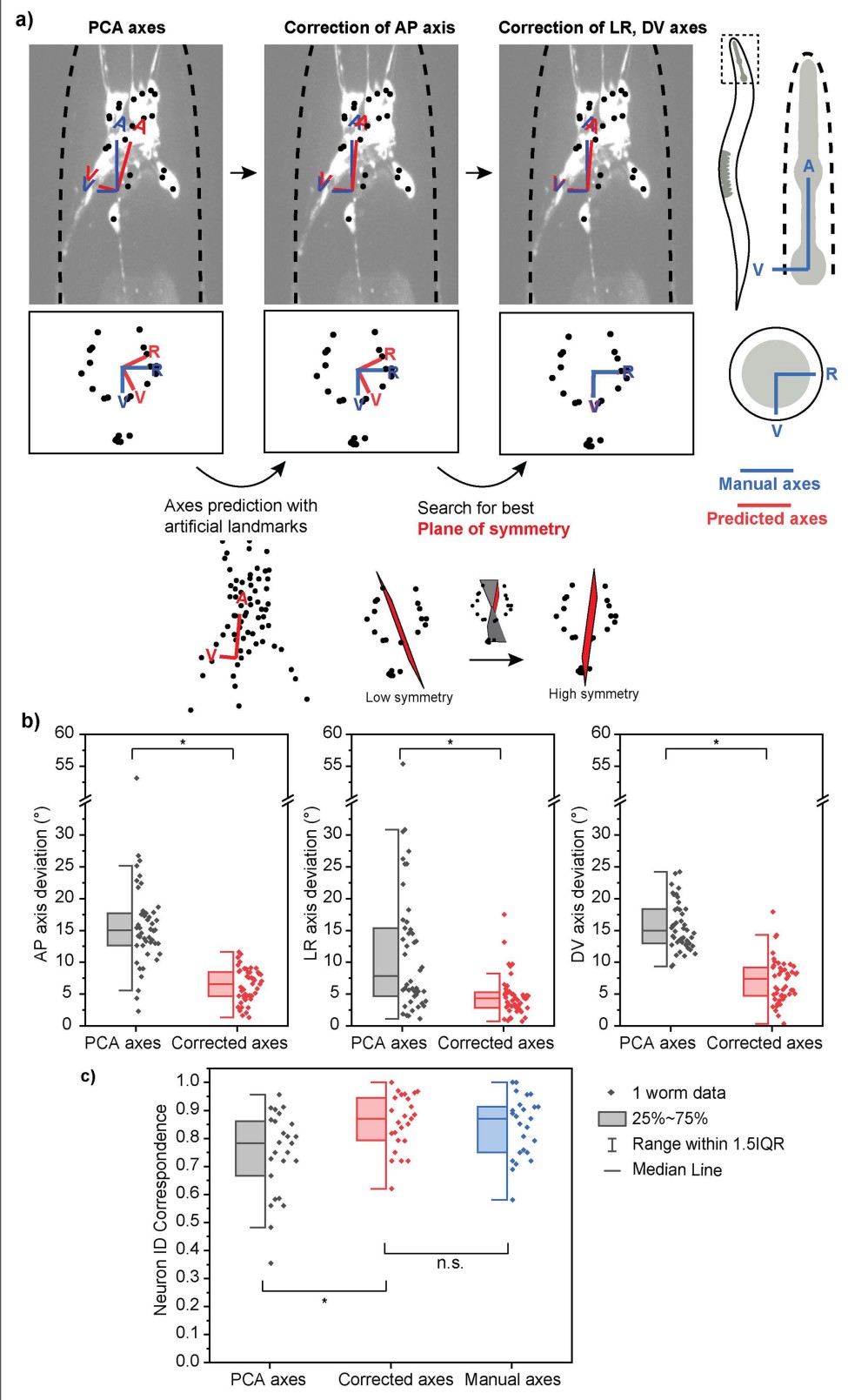

**Figure 2.** Improved method of assigning coordinate axes. (**a**) Coordinate axes for multi-cell images generated by principal components analysis (PCA) alone are not accurate. A two-step correction process is implemented: correction of the AP axis by using natural landmarks and correction of LR, DV axes by searching for the best plane of symmetry. (**b**) The corrected axes are more accurate than the previous axes generated by PCA alone as they

*Figure 2 continued on next page*

*Figure 2 continued*

show decreased angle deviations from the ground truth axes for all three coordinate axes. (**c**) Corrected axes result in a higher and comparable neuron ID accuracy (correspondence to manual cell annotations) when compared with PCA predicted axes and ground truth axes, respectively. Best single prediction results are reported. Two-sample *t*-tests were performed for statistical analysis. The asterisk symbol denotes a significance level of $p<0.05$.

The online version of this article includes the following source data and figure supplement(s) for figure 2:

**Source data 1.** Angle Deviations of PCA and Corrected Axes.

**Source data 2.** Neuron ID Accuracy of PCA, Corrected, and Manual Axes.

**Figure supplement 1.** Neuron ID accuracy no longer depends on the axes inaccuracy after axes correction.

## The atlas's data specificity is important for high neuron identification accuracy

One of the most important requirements for accurate neuron labeling using CRF_ID 2.0 is the availability of an accurate atlas, which serves as a reference map of the stereotypical and probable positions of the neurons in the animal. To characterize the extent to which atlases influence the accuracy of neuron identification and to provide practical guidance on which atlas to use, we evaluated the performance of CRF_ID 2.0 for several possible atlases. In our previous work, we demonstrated that for whole-brain images a data-driven atlas results in higher prediction accuracy (*Chaudhary et al., 2021*). Here, we tested whether the same holds true for multi-cell applications. We characterized several different atlases for predicting neuron identities in worms expressing *glr-1p::NLS-mCherry-NLS* transgene (*Figure 3a and b*). The first atlas is derived from electron microscopy data in the OpenWorm project, which provides the three-dimensional coordinates of neurons in a model adult hermaphrodite *C. elegans* (*Szigeti et al., 2014*). The second atlas is the NeuroPAL atlas, which is the OpenWorm atlas updated with 9 imaging data of the NeuroPAL strain (*Yemini et al., 2021*), and this atlas was reported in our previous work (*Chaudhary et al., 2021*). Several other NeuroPAL atlases from different data sources (*Skuhersky et al., 2022*; *Yemini et al., 2021*) were considered, and the atlas that resulted in the highest neuron ID correspondence was selected (*Figure 3—figure supplement 2*). The other atlases were derived from fluorescence imaging data of *glr-1p::NLS-mCherry-NLS* strain, the same strain of neuron ID interest. Several different versions of the *glr-1* atlas, containing different numbers of datasets, were created to characterize the effect of dataset size on atlas performance.

We examined the effect of data source in atlas performance. Three factors were observed to be most important in determining an atlas's accuracy: strain specificity, mode of data acquisition or imaging conditions, and the number of datasets in the atlas. The neuron annotation accuracy was lowest with the OpenWorm atlas, which scores the poorest in all three factors (*Figure 3c*). This is likely due to the fact that the OpenWorm data are derived from a strain of different genotypical background from the *glr-1p::NLS-mCherry-NLS* strain, and more importantly, data acquisition from electron microscopy and the fluorescence volumetric imaging distort the anatomy differently. In addition, the OpenWorm atlas is based on a single dataset, which does not capture the variability of neuronal positions. A slightly higher accuracy was achieved by using the atlas built on NeuroPAL data (*Figure 3c*). Note that the genotypes of the model training data (the NeuroPAL set) and that of the test set (*glr-1p::NLS-mCherry-NLS*) are still different. The difference between the *glr-1p::NLS-mCherry-NLS* and NeuroPAL strains is significant because, in addition to possible anatomical differences from the genetic make-up, there are neuron pairs in the *glr-1p::NLS-mCherry-NLS* strain that were not updated in the NeuroPAL atlas due to variable expressions of the transgenes used in the NeuroPAL strain and the difficulty of manually annotating all the neurons in the whole-brain images. In fact, 8 (AVBL/R, RIAR, RIGL, RIS, SMDDL/R, SMDVL) out of 37 candidate neurons are missing in the NeuroPAL atlas, which means 40% of the pairwise relationships of neurons expressing the *glr-1p::NLS-mCherry-NLS* transgene were not augmented with the NeuroPAL data but were assigned the default values from the OpenWorm atlas. Further, the imaging conditions for the training and test sets are not entirely comparable because the NeuroPAL images were acquired with a lower z resolution. Such a difference in the imaging condition can lead to differences in segmentation of cell centroids and affect the angular relationships between cells, and thus lowering the accuracy of using the atlas for cell ID prediction. Unlike the OpenWorm atlas, however, the NeuroPAL atlas contains a statistical distribution of neuronal positions from nine datasets, which provides a more accurate representation of the neuronal positions than that from a

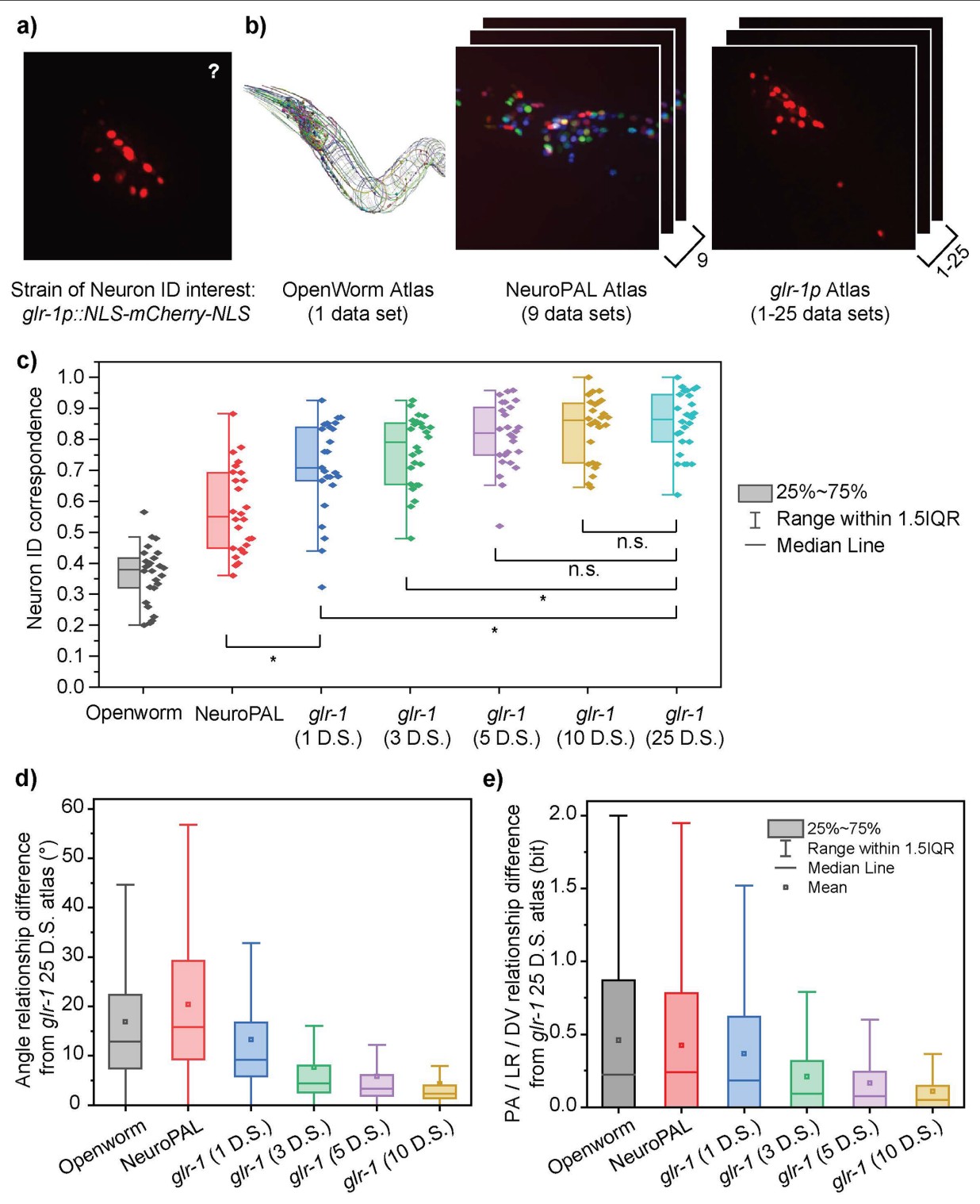

**Figure 3.** Characterizing the importance of data-specific atlases. (**a, b**) Several example atlases (**b**) are compared on their performance in neuron ID prediction on the *glr-1p::NLS-mCherry-NLS* multi-cell images (**a**). (**c**) The neuron ID accuracy (correspondence to manual cell annotations) depends greatly on the atlas used. Each data point represents the cell cluster from one animal (n = 26). Best single prediction results are reported. Two-sample *t*-tests were performed for statistical analysis. The asterisk symbol denotes a significance level of p<0.05. (**d, e**) Difference of each atlas from the most accurate available atlas (*glr-1* 25 datasets) in terms of pairwise angle relationships (**d**) and PA/LR/DV positional relationships (**e**). All distributions in panels (**d**) and (**e**) had a p-value of <0.0001 for one-sample *t*-test against zero.

*Figure 3 continued on next page*

*Figure 3 continued*

The online version of this article includes the following source data and figure supplement(s) for figure 3:

**Source data 1.** Neuron ID Accuracy by Atlas.

**Source data 2.** Diffrence in Angle Relationships by Atlas.

**Source data 3.** Difference in PA/LR/DV Relationships by Atlas.

**Figure supplement 1.** A more detailed visual representation of the difference of each atlas from the best available atlas (*glr-1* from 25 datasets [D.S.]).

**Figure supplement 2.** Comparison of neuron ID correspondences resulting from additional atlases—atlases driven from NeuroPAL neuron positional data from multiple sources (*Chaudhary et al., 2021Chaudhary et al., 2021*; *Skuhersky et al., 2022*; *Yemini et al., 2021*) in red compared to other atlases in *Figure 3*.

**Figure supplement 3.** No correlation between the degree of mosaicism (fraction of cells expressed in the worm) and neuron ID correspondence.

single dataset. Therefore, while in the absence of any data-driven reference atlas standard atlas(es) can be used as a starting point, the more strain-specific datasets used to correct and augment the reference atlas, the more accurate the neuron identification prediction would be.

The highest neuron identification correspondence was found with the data-driven atlas, derived from the *glr-1p::NLS-mCherry-NLS* strain, the same strain that is of interest (*Figure 3c*). The high accuracy can be attributed to the fact that both the strain (thus presumably the anatomy) and the imaging conditions are matching the test dataset. It is notable that even the *glr-1* atlas that is derived from a single dataset performed better than the NeuroPAL atlas containing nine datasets (*Figure 3c*). This implies that the matching strain type and the imaging conditions play a more important role than the sheer number of datasets in the quality of the atlas measured by the cell ID correspondence. Note that we observed no correlation between the degree of mosaicism and neuron ID correspondence (*Figure 3—figure supplement 3*).

We also examined the effect of sample size in atlas building, in addition to the matching strain type and imaging conditions. Atlases should be derived from a sufficiently large sample size to capture the variability within the dataset. Because the neuronal positions are highly variable, an accurate atlas should contain data from a sufficient number of training samples to account for the positional variability of the neurons. As seen with the OpenWorm atlas, an atlas based on only one sample does not contain any statistical information on positional variability, so it performs poorly against testing samples whose neuronal positions do not match those in the atlas well. *Figure 3c* demonstrates that the average correspondence increased with the number of 'ground truth' datasets used to construct the atlas. We referred to the manually annotated cell ID in the training samples as ground-truth. While the *glr-1* atlas constructed using 25 datasets had the highest overall accuracy, the *glr-1* atlases containing 5–10 datasets are statistically indistinguishable in their performance. We observe that the saturation of information is achieved at 5–10 datasets for the *glr-1* case, given that the select datasets exhibit reliable gene expressions to provide statistically good sample sizes for all neuron candidates. This indicates that an atlas derived from 10 well-curated datasets may be sufficient for CRF_ID 2.0. In general, the performance of atlas models would depend on the natural variabilities of the neuronal anatomy and experimental noises, best determined empirically for each strain.

We also compared the differences of the atlases against the best performing atlas (the *glr-1* atlas derived from 25 datasets) as a benchmark. We found that the results correlate well with the trend observed for neuron correspondence (*Figure 3c–e*). The neuron ID correspondence increased with similarity to the *glr-1* atlas with 25 datasets, as defined by the smaller differences in the angular and PA, LR, DV relationships (*Figure 3d and e*; *Figure 3—figure supplement 1*). Interestingly NeuroPAL atlas displayed the highest difference in angular relationship; this is likely due to the difference in the imaging condition for NeuroPAL, in which the fluorescent images were down-sampled in the z direction (*Yemini et al., 2021*). This would cause the neuronal locations to become more discretized along the z axis, which can distort the angular relationships more than the binary relationships. Overall, the results demonstrate that for optimal CRF_ID 2.0 accuracy, it is important to use the atlas derived from data specific to the subject of interest for neuron identification.

## The automated cell annotation accuracy is comparable to manual cell annotation accuracy

In evaluating the performance of CRF_ID 2.0 as an automated cell annotation method, the most important criterion is the accuracy of prediction. It should be noted that we defined accuracy as the correspondence to human annotations (which is how cell ID has been traditionally done), while being cognizant that human annotators do not always provide the absolute ground truth. For this study, the 'ground truth' was established as the cell labels from the consensus of three annotators, which means

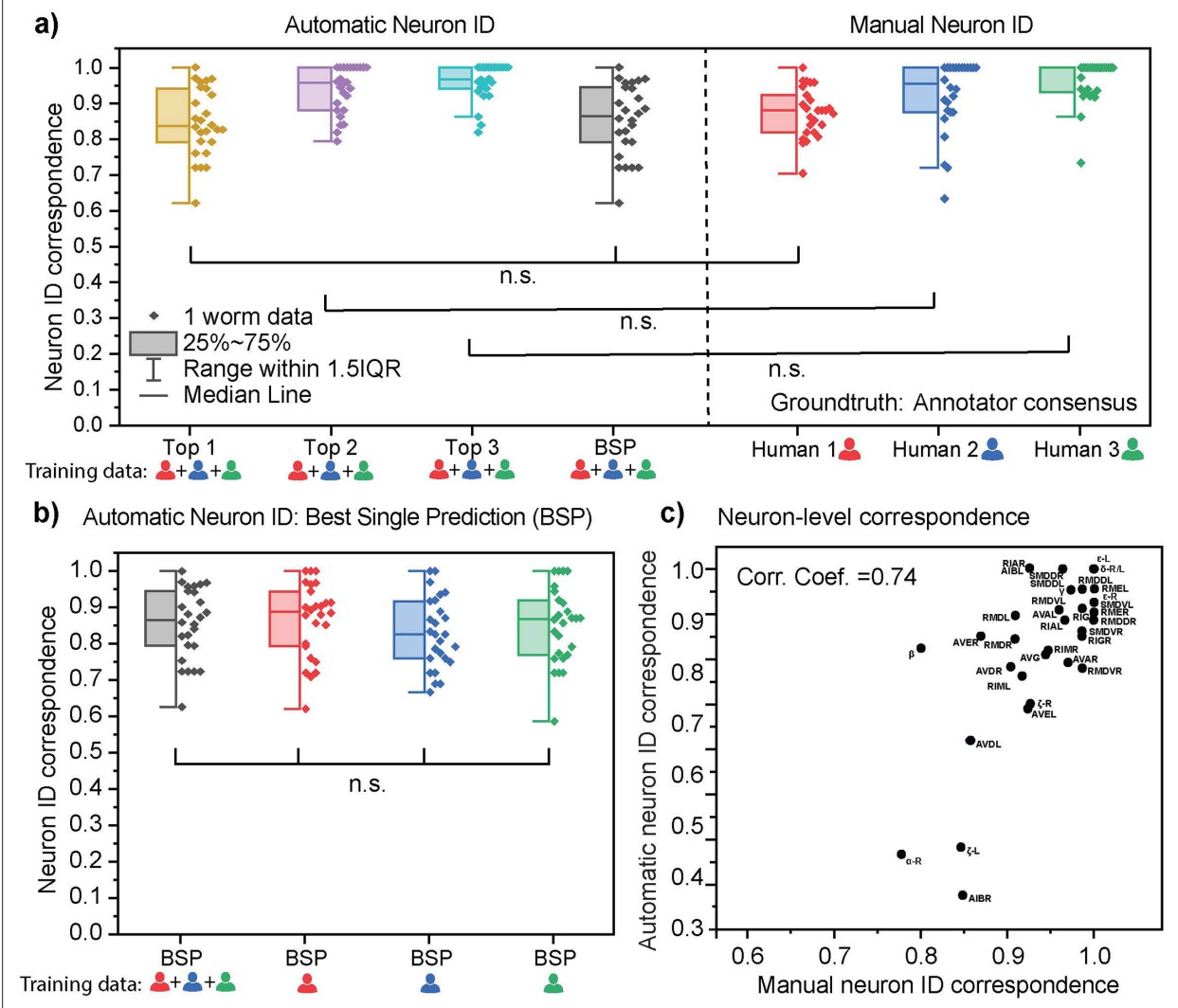

**Figure 4.** Characterization of neuron identification accuracy using CRF_ID 2.0. (**a**) Side-by-side comparison of automated neuron ID accuracy and manual neuron ID accuracy. Each datapoint represents the cell cluster from an animal. For automated neuron ID, top 1, 2, 3 results are from an iterative operation of the CRF_ID algorithm, and the best single prediction (BSP) results are from a single run. The atlas is a compiled data from three different annotators. The ground truth labels are defined by the consensus of the three annotators. Two-sample *t*-tests were performed for statistical analysis (n=26). (**b**) No significant differences in best single prediction accuracies are found when using atlases derived from data annotated by different annotators. One-way ANOVA was performed for statistical analysis. (**c**) There is a positive correlation between the automatic and manual neuron ID accuracy of each neuron.

The online version of this article includes the following source data and figure supplement(s) for figure 4:

**Source data 1.** Neuron ID Accuracy of CRF_ID and Manual Annotations.

**Source data 2.** CRF_ID Accuracy of Atlases Constructed by Different Annotators.

**Source data 3.** Correlation between Automatic and Manual Neuron ID Accuracy.

**Figure supplement 1.** Characterization of manual annotations.

that the ground truth label of a cell is one that has been agreed by at least two annotators. Although there were slight inconsistencies in labels among the three annotators, they had high degrees of correspondence with each other with an average of at least 80% (*Figure 4—figure supplement 1*). The annotations from all three annotators were used as training data, but the accuracy did not decrease to a statistically significant level even when the data from a single annotator were used as the training set (*Figure 4b*). It should be noted that all accuracies reported in this work are based on leave-one-out cross validation, in which the test sample is excluded from the training set.

Because the absolute ground truth labels are often unknown and unattainable even for human annotators, our algorithm provides ranked multiple alternative labels in addition to the best single prediction. These top ranked labels are generated by iteratively running the annotation algorithm with a randomized set of a specific number of neurons in the candidate list removed, similar to our prior approach with building the whole-brain atlas (*Chaudhary et al., 2021*). 'Top 3 accuracy' characterizes the fraction of cells for which the top three predictions include the ground truth label. Note that the top 1 prediction may differ from the best single prediction, which is the result of running the annotation algorithm once with the full set of candidate neurons. The accuracy for the best single prediction mode was around 85%, and those for top 1, 2, 3 predictions were around 85, 94, and 96% respectively (*Figure 4a*). This indicates that in cases where there is appreciable uncertainty of the candidate neuron set, the algorithm can assist the users to decide the final label of a cell by narrowing the candidates and knowing the correct label is almost certainly among the three predictions.

To assess whether the accuracy is acceptable, we compared the aforementioned CRF_ID 2.0 accuracies against the accuracies of human annotators (*Figure 4a*). Annotator 1 had the lowest accuracy, which was statically comparable to the best single prediction and top 1 prediction results from CRF_ID 2.0. Annotators 2 and 3 had higher accuracies, resembling the distributions for top 2 and top 3 prediction results, respectively. Thus, the automated neuron identification using CRF_ID does not come at a loss in accuracy. Moreover, we observed that the incorrect neuron identifications are not entirely unreasonable. When the accuracies are examined per neuron basis, there was a good correlation between the automated and manual neuron ID accuracies. The neurons that were more often incorrectly predicted by the CRF_ID 2.0 algorithm were more likely the ones on which human annotators disagreed with each other (*Figure 4c*). On the other hand, neurons that were 'easy' for human annotators to identify were also more likely to be correctly predicted by the CRF_ID 2.0 algorithm.

## The multi-cell neuron ID is useful for *in vivo* gene expression analysis

To further characterize our multi-cell neuron identification method, we applied it to a problem of biological interest. Although the method has potential uses in any application that requires cells to be identified in a cell population, including calcium imaging (*Ji et al., 2021*; *Nguyen et al., 2016*) and cell lineage tracing (*Bao et al., 2006*), we demonstrate its application in gene expression analysis as an example. Knowing the expression of a particular gene is important for understanding the genetic basis of neuronal function, and it can be studied by examining the expression of mRNA (*Spencer et al., 2011*; *Taylor et al., 2021*), and expression reporters (*Kuroyanagi et al., 2010*). While CeNGEN's mRNA expression data may be an accurate reference for gene expression (*Taylor et al., 2021*), fluorescent reporters allow for *in vivo* monitoring of gene expression on an individual cell basis. This will facilitate studies on changes in gene expression due to perturbations or experimental conditions.

In such applications, the need for neuron identification emerges when the gene of interest is expressed in multiple neurons. For such cases, manually annotating the neurons is difficult and time-consuming, and researchers often resorted to measuring the collective expression of all fluorescence (*Richman et al., 2018*; *Sánchez-Blanco and Kim, 2011*). However, neuron-specific gene expression is more informative and facilitates the elucidation of neuronal functions by connecting the ongoing studies with existing knowledge on specific neurons. For this reason, we applied CRF_ID 2.0 to aid the neuron identification in multi-cell gene expression analysis.

We studied the neuron-specific expression of the *glr-1* gene. *glr-1* encodes a homolog of the mammalian AMPA-type glutamate-gated ionotropic receptor subunits GluA1 and GluA2 (*Brockie et al., 2001*), which play important roles in neural plasticity, learning, and memory (*Henley and Wilkinson, 2016*). *glr-1* is known to be expressed in AVA, AVE, AVD, RMDD/V, RMD, RIM, and SMDD/V among other neurons (*Brockie et al., 2001*; *Hart et al., 1995*; *Maricq et al., 1995*; *Taylor*

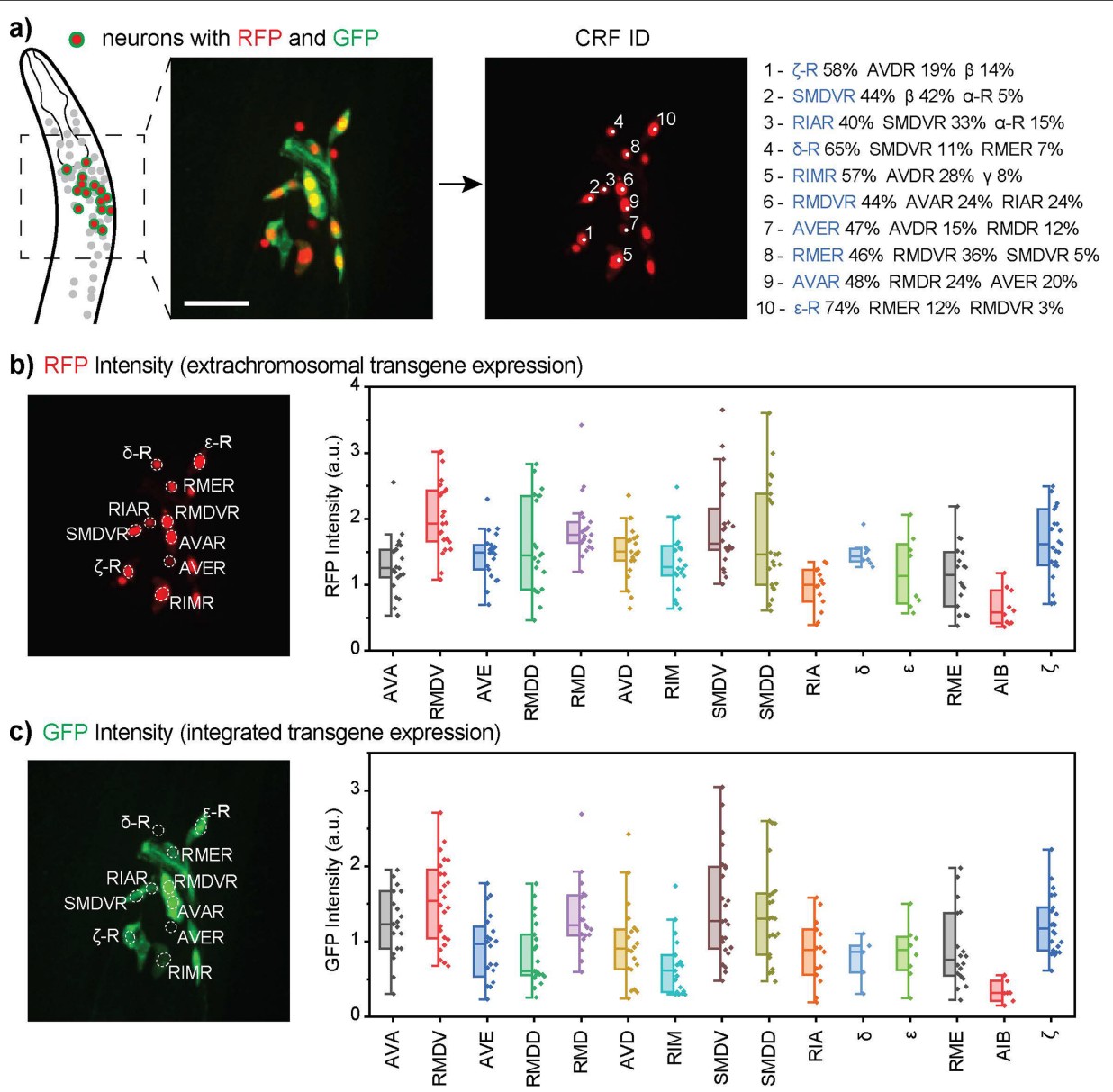

**Figure 5.** Multi-cell neuron identification in *in vivo* gene expression analysis. (**a**) CRF_ID 2.0 facilitates multi-cell annotation by providing top 3 most likely neuron labels for each cell, from which the user makes the final decision. (**b, c**) The example strain contained extrachromosomal (**b**) and integrated reporter transgenes for *glr-1* (**c**). Plotted are the neuron-specific gene expression levels displayed as the normalized fluorescence intensities of selected neurons. The neurons labels on the x-axis are listed in the descending order of single-cell RNA sequencing expression levels reported by CeNGEN (*Taylor et al., 2021*). Box plots indicate median, quartiles and whiskers indicate 1.5 IQR. Data points indicate signals from individual worms (n=30). The images are showing half volume (right side) of the specimen for illustration. a.u.: arbitrary unit.

The online version of this article includes the following source data and figure supplement(s) for figure 5:

**Source data 1.** Neuron-Specific Extrachromosomal Transgene Expression.

**Source data 2.** Neuron-Specific Integrated Transgene Expression.

**Figure supplement 1.** Neuron-specific expressions of the *glr-1* gene for neurons on the 'dim' side of the specimen.

**Figure supplement 2.** High correlation between extrachromosomal (mCherry) and integrated (GFP) transgene expressions.

*et al., 2021*). We imaged a *C. elegans* strain that has two types of reporters, an integrated transgene (*glr-1::gfp*) and an extrachromosomal transgene with nuclear-localized sequences (*glr-1p::NLS-mCherry-NLS*) (*Figure 5a*). In general, the expression of an integrated transgene is more robust and more reflective of the physiological expression because, unlike the expression of an extrachromosomal

transgene, it is more resistant to genetic mosaicism, in which a transgene is not inherited to some cells during cell divisions due to instability of the extrachromosomal DNA (*Frøkjaer-Jensen et al., 2008*).

To quantify expression, we next segmented the neurons from the mCherry image stacks. Because of nuclear localization, the fluorophore signals are more resolved between cells. We then applied CRF_ID 2.0 to the multi-cell point cloud to assign cell identities (*Figure 5a*). The neuron identification results were provided as top three candidate labels for each cell for the user to decide on the final labels. There were about 20–35 fluorescent cells in each volume. The fluorescence intensities of paired neurons were pooled together and then regrouped into two: one on the brighter side closer to the objective and one on the dimmer side farther from the objective. *Figure 5b and c* only report the fluorescent intensities of neurons on the brighter side (the side closer to the imaging objective), but the dimmer side showed a very similar profile (*Figure 5—figure supplement 1*).

The gene expression analysis revealed several insights. First, we did not observe a significant difference in relative expression trends between extrachromosomal (*Figure 5b*) and integrated transgenes (*Figure 5c*). For example, the neurons that had high expression levels for the integrated transgene, such as AVA, RMDV, RMD, SMDV, SMDD, and $\zeta$, also exhibited high expression levels for the extrachromosomal transgene. More specifically, there was a linear correlation between GFP and mCherry intensities (*Figure 5—figure supplement 2*). Second, the extrachromosomal transgene expression did not have particularly more variable expression levels. This suggests that gene expression studies with extrachromosomal transcriptional reporters, which avoid the time-consuming process of gene integration, can still provide robust and meaningful insights into gene expression. Third, gene expressions in neurons were found to be highly variable among individual samples, which was generally expected considering the dynamic nature of gene expressions. The neurons with high average expression levels showed a wider range of variability.

Interestingly, the mRNA profiling data generated by CeNGEN do not correlate well with the results from transgene expressions. In the plots in *Figure 5*, the neuron labels on the x-axis are listed in the order of decreasing expression level according to CeNGEN (*Taylor et al., 2021*). The low correlation is likely due to several reasons. First, there is an innate difference between mRNA expression and transcriptional reporter signals, with the reporters more indicative of the expression of the fusion proteins. Second, the promoter sequence in the transgene, while carefully chosen, could still be different from the native *cis*-regulatory sequence that could include sequences residing in the intron regions or downstream regions all of which may regulate mRNA expression measured by sequencing. Lastly, the cell dissociation process needed for profiling mRNAs of individual neurons may impact gene expression.

## Discussion

In this work, we have upgraded and characterized the original CRF_ID method to accommodate common needs to analyze more diverse types of biological images. While the original method demonstrated high annotation accuracy in *C. elegans* whole-brain images, its performance in multi-cell images, which include all images of neuronal groups that are not whole-brain, was not fully guaranteed. The modified axes assignment method accurately predicts the coordinate axes of *C. elegans* volumetric images regardless of the number of cells expressed and thereby greatly enhances cell identification success. Our characterization of the CRF_ID 2.0 performance in comparison with the atlas and human annotators offers an important benchmark for users interested in using CRF_ID 2.0 to annotate multi-cell images. In addition, we demonstrate its application in transgenic reporter-based gene expression analysis, which inherently invovles multi-cell expression. This work represents a practical advance, with updated features better suited for multi-cell applications, and still applicable to whole-brain images.

One of the distinguishing advantages of CRF_ID 2.0 is its flexibility, which stems from its modular architecture. The fundamental framework has not been changed from CRF_ID 1.0, and therefore the advantages of CRF_ID outlined in the original work apply for CRF_ID 2.0 as well. Compared with registration or deep learning-based methods, in which it is difficult or impossible to adjust the optimization process, the graphical model approach of CRF_ID allows heuristic-based deliberate selection and tuning of the features. This aspect opens the door for many other use cases. For example, for images with distinctive cellular characteristics, the user may add new unary features for characteristics, such as the cell shape and signal intensity, to be optimized. For imaging conditions for which

certain features become less reliable, the user may choose to reduce or remove the weights for those features. Further, the simplicity of building the atlas is another important aspect that enables the incorporation of new images of different strains and imaging conditions into these data-driven custom atlases. If one determines that the anatomy of a particular strain is substantially different from existing atlases (e.g., due to genetic background or rearing conditions), new atlases can be easily constructed using existing atlases as starting points and updating differences from the data. Unlike deep learning methods, which would require high-performance computing and hours of training time, our atlases can be built or updated from existing atlases within seconds with basic arithmetical operations. As such, CRF_ID can be easily applied to biological images of various origins, not restricted to *C. elegans* that display stereotypical cellular characteristics.

## Materials and methods

**Key resources table**

| Reagent type (species) or resource | Designation | Source or reference | Identifiers | Additional information |
|---|---|---|---|---|
| Strain, strain background (*Caenorhabditis elegans*) | ZC3292 | This work | yxEx1701[glr-1p::GCaMP6s, glr-1p::NLS-mCherry-NLS] | Available upon request from Yun Zhang |
| Strain, strain background (*C. elegans*) | ZC3612 | This work | lin-15B&lin-15A(n765) kyIs30[glr-1::GFP, lin-15(+)] X; yxEx1933[glr-1p::NLS-mCherry-NLS] | Available upon request from Yun Zhang |

### Strains

In this study, adult *C. elegans* hermaphrodites were used and cultured using standard procedures (*Brenner, 1974*). Two *C. elegans* transgenic strains were used. All data reported in *Figures 1–4*, including the training data for the *glr-1* atlases, are from images of ZC3292 *yxEx1701*[*glr-1p::GCaMP6s, glr-1p::NLS-mCherry-NLS*]. The gene expression data in *Figure 5* are from images of ZC3612 *lin-15B&lin-15A(n765) kyIs30[glr-1::GFP, lin-15(+)] X; yxEx1933[glr-1p::NLS-mCherry-NLS]*.

### Construction of transgenes and transgenic strains

The *glr-1p::GCaMP6s* plasmid was made by LR recombination (NEB) of a destination vector containing the DNA sequences for GCaMP6s (*Chen et al., 2013*) and for *unc-54* 3'UTR (from a gift plasmid from Andrew Fire) with an entry vector containing a 5.3 kb DNA sequence for the *glr-1* promoter (*Brockie et al., 2001*). The *glr-1p::NLS-mCherry-NLS* plasmid was made by LR recombination of the entry vector containing the *glr-1* promoter with a destination vector (generated using a gift plasmid from Joshua Kaplan) containing a mCherry sequence flanked by two NLSs (nuclear localization sequences) followed by the sequence for *unc-54* 3'UTR constructed using Gibson assembly (Thermo Fisher). Each plasmid was injected at 10 ng/μL to generate ZC3292, and the *glr-1p::NLS-mCherry-NLS* plasmid was injected at 40 ng/μL to generate ZC3612 by following the standard methods of microinjection (*Mello et al., 1991*).

### Imaging data collection

Imaging of the ZC3292 strain was performed using an Andor Spinning Disk confocal system connected with a Nikon Eclipse Ti-E inverted microscope. A 40x oil-immersion objective (N.A.=1.3) was used, and images were recorded using an ANDOR iXon Ultra EMCCD camera. Hermaphrodites at day 1 adult stage were immobilized with sodium azide in the chamber of a microfluidic device that used an AutoMate Scientific ValveBank perfusion system (Berkeley, CA) as the controller (*Chronis et al., 2007*). Volumetric images of the head region were recorded in two channels (green channel using a laser at 488 nm and a filter at 525 nm/50 nm and red channel using a laser at 561 nm and a filter at 617 nm/73 nm). The Z step size is 0.3 μm and the XY resolution is 0.4 μm. The exposure time is 20 ms for both channels.

Imaging of the strain ZC3612 was performed using a Nikon W1 spinning disk confocal system on Nikon Ti2-E microscope with Hamamatsu ORCA-Fusion Gen-III sCMOS camera. A ×60 objective lens with N.A. 1.4 was used. The animals were age-synchronized to day 1 adult stage and chemically immobilized in 20 mM tetramisole. In order to efficiently image straight-headed animals, we loaded the animals into an array-type microfluidic device (*Lee et al., 2014*). 3D stacks of the animal's head

region in red (laser: 561 nm; filter: T605/52 m) and green (laser: 488 nm; filter: ET525/36 m) channels were acquired with a Z step size of 0.3 µm. The exposure times were 10 ms and 50 ms for red and green channels, respectively. The XY resolution of the images was 0.12 µm.

## Manual annotation and atlas construction

Building an atlas requires manually annotated datasets. Three participants separately segmented and annotated cells on raw 3D stacks of 26 worm data. Most of the annotations were done by visually comparing the cell point cloud against reference images. The references include the anatomical features of head neurons, including the positions of the cell bodies and the shape of the neuronal processes, on the WormAtlas website *Altun et al., 2002*, as well as 3D representation of the neuron coordinates on OpenWorm (*Szigeti et al., 2014*). 3D reconstruction of fluorescent neuronal images was also used to determine cell identity. The list of candidate neurons expressing *glr-1* was carefully curated based on known expression patterns (*Brockie et al., 2001*; *Hart et al., 1995*; *Maricq et al., 1995*; *Taylor et al., 2021*) and by analyzing the expression patterns in our own data. The list of annotated neurons is as follows: AIBL/R, AVAL/R, ζ-L/R, AVDL/R, AVEL/R, AVG, α-L/R, β, RIAL/R, RIGL/R, RIML/R, RMDL/R, RMDDL/R, RMEL/R, γ, RMDVL/R, SMDDL/R, SMDVL/R, δ-L/R, ε-L/R. Neuronal expressions with low confidence have been indicated with Greek symbols; we cautiously mention α, β, γ, δ, ε, ζ may correspond to AVJ, M1, RIS, URYD, URYV, and AVB respectively. The annotation of left versus right neurons of a pair follows Individual Neurons List on WormAtlas. The annotation of RIGL versus RIGR follows the annotation shown on WormAtlas here. The data-driven atlas was constructed using the atlas generation codes included in the original CRF_ID. While the human annotators applied the same methods, their annotations of neuron IDs are not identical, which reflected variations commonly observed for manual cell ID annotations. Instead of building an atlas from the consensus labels of three annotators, the labels from different annotators were considered as separate annotations, effectively capturing the statistics of the information from the dataset instead of using majority-vote single label. For multi-cell neuron predictions on the *glr-1* strains, a truncated atlas containing only the above 37 neurons was used to exclude neuron candidates that are irrelevant for prediction.

## CRF_ID 2.0: Improved axes prediction

The new axes prediction method consists of two parts: AP axis correction and LR/DV axes correction. For AP axis correction, the point cloud was artificially enlarged by including naturally fluorescent landmarks in the animal's body. The autofluorescence signals were segmented by thresholding local maxima from the image after Gaussian filtering. The threshold value of 99.85 was determined experimentally and can be tuned for different imaging conditions. Then, PCA was applied to the point cloud to find the AP axis and the initial LR and DV axes.

The algorithm for LR/DV axes correction takes the initial LR and DV axes and cell point cloud as inputs and outputs the corrected LR and DV axes. It searches for the plane/axes pair that divides the point cloud into two sides with the highest symmetry across the plane. First, it finds a range of planes that satisfy the following constraints: orthogonality to the AP axis, within the angle of 20° from the initial axis, and an increment of 1°. The initial axis is the LR axis that was generated by PCA in the previous step. By iteratively testing each plane, the one that results in highest symmetry was found. Symmetry was quantified as the inverse of the deviation from perfect symmetry. The deviation was calculated by reflecting each point with respect to the plane and calculating the distance of the reflected point to the closest neighboring point. The distance was calculated for all points in the point cloud and the average of the lowest seven distances was used for the given plane. The threshold value of 7 was derived from the expectation of at least seven left/right pairs of neurons in the *glr-1* strain, but this threshold can be tuned depending on the characteristics of the strain or the images. Higher symmetry with respect to a plane would result in a lower distance value. The plane that resulted in the lowest distance was assigned the final LR axis, and the DV axis is automatically determined by orthogonality to the first two axes. The average run time for each point cloud was under 1 min.

## CRF_ID 2.0: Evaluation of accuracy

Because the absolute ground truth neuron labels for the cells are not available, we defined the annotation accuracy as correspondence to the consensus labels of three human annotations. The

consensus was established as the label that has been agreed by at least two annotators. Cells whose identities were differently annotated by all three annotators, which account for around 6% of all cells, were omitted from the accuracy calculation. Also, all reported correspondence values are results of leave-one-out cross-validations, meaning the atlas used to test the accuracy on a specific worm did not include that specific dataset. Thus, there are 26 different versions of the *glr-1* 25 dataset atlas, one for each dataset exclusion. For example, in *Figure 3c*, a correspondence value of 0.8 indicates that 80% of the cells in the dataset were assigned the neuron label that matches the consensus label, and the atlas used is driven from manual annotations of 25 other worm datasets. The correspondences reported in *Figures 2 and 3* are from results of running the prediction algorithm once in the best single prediction mode. *Figure 4* additionally reports top 1, 2, 3 results from 100 iterations while randomly removing neuron labels from the candidate list.

## Gene expression analysis

To extract the cell-specific fluorescent signals, we first used the automatic segmentation tool to segment the cells in a total of 27 image stacks of ZC3612. The segmentation was done on the nuclear-localized mCherry signals, and GFP intensities were extracted from the same region. The quality of the segmentation was visually inspected to eliminate false positives. Then, iterative neuron ID predictions were performed on the segmented red channel images. The resulting top 3 candidates were reviewed for each cell, and the final neuron label was selected based on human judgment. Then, a series of data processing steps were necessary to best represent the cell-specific gene expression data from different animals. First, we extracted the mCherry and GFP signal intensities by averaging the intensities of the brightest 100 pixels in the segmented masks, which had around 500 pixels on average. Because mCherry and GFP expressions are driven by separate transgenes, mCherry expression does not guarantee GFP expression. Thus, GFP expressions that were deemed absent were eliminated during data curation. Second, because the expression levels are variable among animals, the fluorescence intensities were normalized for each animal by dividing by the average intensity of all neurons in the animal. Lastly, we compared the intensities of cells only on the same side of the animal because the side of the animal closer to the objective lens was generally brighter due to light scattering through the biological tissues. The reported values in *Figure 5* are the intensities on the brighter side of the animal.

## Statistical analysis

Statistical analyses of the data were performed using Paired Comparisons App in OriginPro 2020. The asterisk symbol denotes a significance level of $p<0.05$. Nonsignificantly different comparisons are denoted non-significant (n.s.).

## Acknowledgements

The authors acknowledge the funding support of the US NIH (R01MH130064, R01NS115484) to HL and YZ and the US NSF (1764406) to HL. Some nematode strains used in this work were provided by the Caenorhabditis Genetics Center (CGC), which is funded by the NIH Office of Research Infrastructure Programs (P40 OD010440) National Center for Research Resources.

## Additional information

### Funding

| Funder | Grant reference number | Author |
|---|---|---|
| National Institutes of Health | R01MH130064 | Yun Zhang Hang Lu |
| National Institutes of Health | R01NS115484 | Yun Zhang Hang Lu |
| National Science Foundation | 1764406 | Hang Lu |

| Funder | Grant reference number | Author |
|---|---|---|

The funders had no role in study design, data collection and interpretation, or the decision to submit the work for publication.

## Author contributions

Hyun Jee Lee, Conceptualization, Data curation, Software, Validation, Investigation, Visualization, Methodology, Writing - original draft, Writing - review and editing; Jingting Liang, Data curation, Validation, Investigation, Writing - review and editing; Shivesh Chaudhary, Conceptualization, Software, Methodology; Sihoon Moon, Conceptualization, Software, Writing - review and editing; Zikai Yu, Software, Methodology, Writing - review and editing; Taihong Wu, He Liu, Myung-Kyu Choi, Data curation; Yun Zhang, Conceptualization, Resources, Supervision, Funding acquisition, Project administration, Writing - review and editing; Hang Lu, Conceptualization, Resources, Supervision, Funding acquisition, Methodology, Project administration, Writing - review and editing

## Author ORCIDs

Hyun Jee Lee http://orcid.org/0000-0001-9662-2063
Jingting Liang http://orcid.org/0009-0004-4284-257X
Shivesh Chaudhary https://orcid.org/0000-0002-1928-0933
Sihoon Moon https://orcid.org/0000-0003-4540-9443
Zikai Yu http://orcid.org/0009-0004-1851-3493
Taihong Wu https://orcid.org/0000-0002-9760-6978
He Liu https://orcid.org/0000-0001-9418-9171
Yun Zhang https://orcid.org/0000-0002-7631-858X
Hang Lu https://orcid.org/0000-0002-6881-660X

Reviewer #1 (Public Review): https://doi.org/10.7554/eLife.89050.4.sa1
Reviewer #2 (Public Review): https://doi.org/10.7554/eLife.89050.4.sa2
Author response https://doi.org/10.7554/eLife.89050.4.sa3

# Additional files

## Supplementary files

MDAR checklist

## Data availability

CRF_ID 2.0 can be accessed at https://github.com/lu-lab/CRF-Cell-ID-2.0 (copy archived at *lu-lab, 2023*). This repository contains all components of the framework and the atlases produced and compared in this work. Experimental data used to generate the figures are provided as separate data source files.

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
