## [Editor Report · eLife Assessment]

This Research Advance describes a **valuable** image analysis method to identify individual neurons within a ‎population of fluorescently labeled cells in the nematode *C. elegans*. The findings are **solid** and the method succeeds to identify cells with high precision. The method will be be of interest to the *C. elegans* research community.

---

## [Referee Report · Reviewer #1 (Public Review)]

In this paper, the authors developed an image analysis pipeline to automatically identify individual ‎‎neurons within a population of fluorescently tagged neurons. This application is optimized to deal with ‎‎multi-cell analysis and builds on a previous software version, developed by the same team, to resolve ‎‎individual neurons from whole-brain imaging stacks. Using advanced statistical approaches and ‎‎several heuristics tailored for *C. elegans* anatomy, the method successfully identifies individual ‎‎neurons with a fairly high accuracy. Thus, while specific to *C. elegans*, this method can ‎become ‎instrumental for a variety of research directions such as in-vivo single-cell gene expression ‎analysis ‎and calcium-based neural activity studies.‎

---

## [Referee Report · Reviewer #2 (Public Review)]

The authors succeed in generalizing the pre-alignment procedure for their cell identification method to allow it to work effectively on data with only small subsets of cells labeled. They convincingly show that their extension accurately identifies head angle, based on finding auto florescent tissue and looking for a symmetric l/r axis. Their demonstrated method works to allow the identification of a particular subset of neurons. Their approach should be a useful one for researchers wishing to identify subsets of head neurons in *C. elegans*, and the ideas might be useful elsewhere.

The authors also assess the relative usefulness of several atlases for making identity predictions. They attempt to give some additional general insights on what makes a good atlas, and clearly demonstrate the value of more data. Some insights seem less clear as available data do not allow for experiments that cleanly decouple: (1) the number of examples in the atlas; (2) the completeness of the atlas; and (3) the match in strain and imaging modality discussed. In the presented experiments the custom atlas, besides the strain and imaging modality congruence discussed is also the only complete atlas with more than one example. The main neuroPAL atlas is an imperfect stand-in since a significant fraction of cells could not be identified in these data sets, making it a 60/40 mix of Openworm and a hypothetical perfect neuroPAL comparison. The alternate neuroPal atlases shown in supplemental figure 4 are complete but provide only one point cloud.

It is striking that in the best available apples to apples match the single data set glr-1 atlas produces qualitatively better results than the single (complete) neuroPAL atlas. This is a clear performance advantage given the ground truth. This is as good an evaluation as is possible given current data however given the inexact nature of assigning ground truth identities I think it is difficult from results to tease out if this is due to strain, imaging conditions or systematically different identifications of cells from different sources.

The experiments do usefully explore the volume of data needed. Though generalization to other arbitrary cell subsets remains to be shown the insight is useful for future atlas building that for the specific (small) set of cells labeled in the experiments 5-10 examples is sufficient to build an accurate atlas.

---

## [Author Response]

The following is the authors’ response to the previous reviews.

**Public Reviews:**

**Reviewer #1 (Public Review):**
In this paper, the authors developed an image analysis pipeline to automacally idenfy individual neurons within a populaon of fluorescently tagged neurons. This applicaon is opmized to deal with mul-cell analysis and builds on a previous soware version, developed by the same team, to resolve individual neurons from whole-brain imaging stacks. Using advanced stascal approaches and several heuriscs tailored for *C. elegans* anatomy, the method successfully idenfies individual neurons with a fairly high accuracy. Thus, while specific to *C. elegans*, this method can become instrumental for a variety of research direcons such as in-vivo single-cell gene expression analysis and calcium-based neural acvity studies.

Thank you.

**Reviewer #2 (Public Review):**
The authors succeed in generalizing the pre-alignment procedure for their cell idenficaon method to allow it to work effecvely on data with only small subsets of cells labeled. They convincingly show that their extension accurately idenfies head angle, based on finding auto florescent ssue and looking for a symmetric l/r axis. They demonstrate method works to allow the idenficaon of a parcular subset of neurons. Their approach should be a useful one for researchers wishing to idenfy subsets of head neurons in *C. elegans*, and the ideas might be useful elsewhere.The authors also assess the relave usefulness of several atlases for making identy predicons. They atempt to give some addional general insights on what makes a good atlas, but here insights seem less clear as available data does not allow for experiments that cleanly decouple: 1. the number of examples in the atlas 2. the completeness of the atlas. and 3. the match in strain and imaging modality discussed. In the presented experiments the custom atlas, besides the strain and imaging modality mismatches discussed is also the only complete atlas with more than one example. The neuroPAL atlas, is an imperfect stand in, since a significant fracon of cells could not be idenfied in these data sets, making it a 60/40 mix of Openworm and a hypothecal perfect neuroPAL comparison. This waters down general insights since it is unclear if the performance is driven by strain/imaging modality or these difficules creang a complete neuroPal atlas. The experiments do usefully explore the volume of data needed. Though generalizaon remains to be shown the insight is useful for future atlas building that for the specific (small) set of cells labeled in the experiments 5-10 examples is sufficient to build aaccurate atlas.

The reviewer brings up an interesting point. As the reviewer noted, given the imperfection of the datasets (ours and others’), it is possible that artifacts from incomplete atlases can interfere with the assessment of the performances of different atlases. To address this, as the reviewer suggested, we have searched the literature and found two sets of data that give specific coordinates of identified neurons (both using NeuroPAL). We compared the performance of the atlases derived from these datasets to the strain-specific atlases, and the original conclusion stands. Details are now included in the revised manuscript (Figure 3- figure supplement 2).

**Recommendaons for the authors:**

**Reviewer #1 (Recommendaons For The Authors):**
I appreciate the new mosaic analysis (Fig. 3 -figure suppl 2). Please fix the y-axis ck label that I believe should be 0.8 (instead of 0.9).

We thank the reviewer for spotting the typo. We have fixed the error.

**Reviewer #2 (Recommendaons For The Authors):Though I'm not familiar with the exact quality of GT labels in available neuroPAL data I know increasing volumes of published data is available. Comparison with a complete neuroPAL atlas, and a similar assessment on atlas size as made with the custom atlas would to my mind qualitavely increase the general insights on atlas construcon.

We thank the reviewer for the insightful suggestion. We have newly constructed several other NeuroPAL atlases by incorporating neuron positional data from two other published data: [Yemini E. et al. NeuroPAL: A Multicolor Atlas for Whole-Brain Neuronal Identification in *C. elegans*. Cell. 2021 Jan 7;184(1):272-288.e11] and [Skuhersky, M. et al. Toward a more accurate 3D atlas of *C. elegans* neurons. BMC Bioinformatics 23, 195 (2022)].

Interestingly, we found that the two new atlases (NP-Yemini and NP-Skuhersky) have significantly different values of PA, LR, DV, and angle relationships for certain cells compared to the OpenWorm and glr-1 atlases. For example, in both the NP atlases, SMDD is labeled as being anterior to AIB, which is the opposite of the SMDD-AIB relationship in the glr-1 atlas.

Because this relationship (and other similar cases) were missing in our original NeuroPAL atlas (NP-Chaudhary), the addition of these two NeuroPAL datasets to our NeuroPAL atlas dramatically changed the atlas. As a result, incorporating the published data sets into the NeuroPAL atlas (NP-all) actually decreased the average prediction accuracy to 44%, while the average accuracy of original NeuroPAL atlas (NP-Chaudhary) was 57%. The atlas based on the Yemini et al. data alone (NP-Yemini) had 43% accuracy, and the atlas based on the Skuhersky et al. data alone (NP-Skuhersky) had 38% accuracy.

For the rest of our analysis, we focused on comparing the NeuroPAL atlas that resulted in the highest accuracy against other atlases in figure 3 (NP-Chaudhary). Therefore, we have added Figure 3- figure supplement 2 and the following sentence in the discussion. “Several other NeuroPAL atlases from different data sources were considered, and the atlas that resulted in the highest neuron ID correspondence was selected (Figure 3- figure supplement 2).”

**Author response image 1. sa3fig1:** Figure3- figure supplement 2. Comparison of neuron ID correspondences resulng from addional atlases- atlases driven from NeuroPAL neuron posional data from mulple sources (Chaudhary et al., Yemini et al., and Skuhersky et al.) in red compared to other atlases in Figure 3. Two sample t-tests were performed for stascal analysis. The asterisk symbol denotes a significance level of p<0.05, and n.s. denotes no significance. OW: atlas driven by data from OpenWorm project, NP-source: NeuroPAL atlas driven by data from the source. NP-Chaudhary atlas corresponds to NeuroPAL atlas in Figure 3.

80% agreement among manual idenficaons seems low to me for a relavely small, (mostly) known set of cells, which seems to cast into doubt ground truth idenes based on a best 2 out of 3 vote. The authors menon 3% of cell idenes had total disagreement and were excluded, what were the fracon unanimous and 2/3? Are there any further insights about what limited human performance in the context of this parcular idenficaon task?

We closely looked into the manual annotation data. The fraction of cells in unanimous, two thirds, and no agreement are approximately 74%, 20%, and 6%, respectively. We made the corresponding change in the manuscript from 3% to 6%. Indeed, we identified certain patterns in labels that were more likely to be disagreed upon. First, cells in close proximity to each other, such as AVE and RMD, were often switched from annotator to annotator. Second, cells in the posterior part of the cluster, such as RIM, AVD, AVB, were more variable in positions, so their identities were not clear at times. Third, annotators were more likely to disagree on cells whose expressions are rare and low, and these include AIB, AVJ, and M1. These observations agree with our results in figure 4c.